# Error-Aware Imitation Learning from Teleoperation Data for Mobile Manipulation

**Josiah Wong, Albert Tung, Andrey Kurenkov, Ajay Mandlekar,**
**Li Fei-Fei, Silvio Savarese, Roberto Martín-Martín**

Stanford University

**Abstract:** In mobile manipulation (MM), robots can both navigate within and interact with their environment and are thus able to complete many more tasks than robots only capable of navigation or manipulation. In this work, we explore how to apply imitation learning (IL) to learn continuous visuo-motor policies for MM tasks. Much prior work has shown that IL can train visuo-motor policies for either manipulation or navigation domains, but few works have applied IL to the MM domain. Doing this is challenging for two reasons: on the data side, current interfaces make collecting high-quality human demonstrations difficult, and on the learning side, policies trained on limited data can suffer from covariate shift when deployed. To address these problems, we first propose MOBILE MANIPULATION ROBOTURK (MOMART), a novel teleoperation framework allowing simultaneous navigation and manipulation of mobile manipulators, and collect a first-of-its-kind large scale dataset in a realistic simulated kitchen setting. We then propose a learned error detection system to address covariate shift by detecting when an agent is in a potential failure state. We train performant IL policies and error detectors from this data, and achieve over 45% task success rate and 85% error detection success rate across multiple multi-stage tasks when trained on expert data. Additional results and video at https://sites.google.com/view/il-for-mm/home.

**Keywords:** Mobile Manipulation, Imitation Learning, Error Detection

## 1 Introduction

In mobile manipulation (MM), robots combine interaction and locomotion capabilities to substantially extend the depth and breadth of tasks they can perform compared to static manipulation [1, 2, 3, 4]. For example, in household environments MM agents can perform tasks such as cleaning up the dining table and bringing the dishes to the dishwasher. Data-driven paradigms such as reinforcement learning are attractive for MM since they enable such agents to learn directly from raw sensory observations, so that they can function in a wide variety of households. However, due to the many possible interactions that are possible for the agent (looking, navigating, and manipulating various parts of the household) the state space for MM tasks is vast, which imposes a difficult exploration burden for agents that learn autonomously. Learning from human demonstrations has been extremely effective in addressing the burden of exploration in static manipulation settings. Can we apply the same paradigm for mobile manipulation?

Unfortunately, the vast state space in MM has inhibited data collection from humans due to the difficulty of annotating substantial portions of the state space with good actions. Prior work has avoided this issue by instrumenting the environment to ease the state coverage burden [5, 6, 7, 8]. However, even if there were a way to collect full human demonstrations without instrumentation, learning from such datasets would also pose a challenge due to covariate shift [9], where trained policies visit states not covered by the demonstrations – an outcome that is more likely in MM. In this paper, we address the problem of collecting and learning from human demonstrations in mobile manipulation settings via two key solutions: a novel system to collect MM demonstrations, and an algorithmic framework that can detect when an agent is suffering from covariate shift.

5th Conference on Robot Learning (CoRL 2021), London, UK.

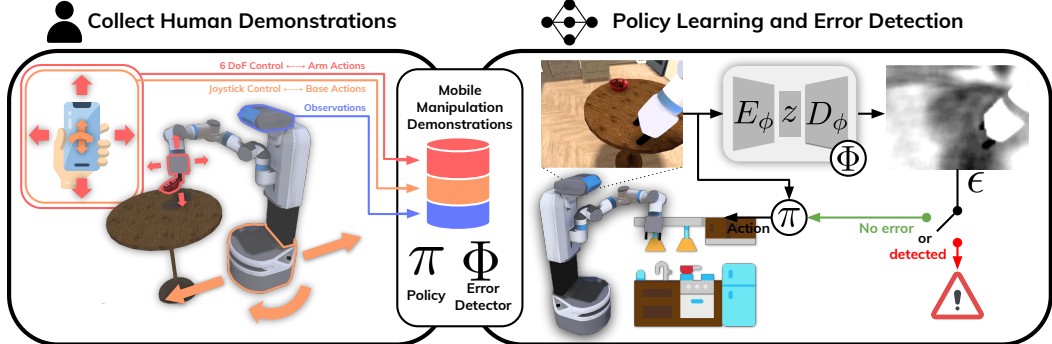

Figure 1: **Leveraging Human Demonstrations for Mobile Manipulation.** Humans have strong intuitions about how to move and interact based on visual data. We leverage their knowledge by first collecting human demonstrations teleoperating a mobile manipulator using MOMART *(left)*, and proceed to train imitation learning policies and error detectors from the resulting dataset *(middle)*. During rollouts, the policy executes actions continuously while the error detector simultaneously checks if the current visual state can be reconstructed. A sufficiently irregular state provokes the error detector to intervene, either resetting the agent to well-known configuration or immediately terminating to prevent erratic policy behavior *(right)*.

First, we present MOBILE MANIPULATION ROBOTURK (MOMART), a novel teleoperation system that allows humans to remotely control mobile manipulators in a natural and easy manner. Operators control the robot's motion using their smartphone to provide real-time navigation and manipulation commands simultaneously. Unlike prior works on IL for MM which leverage privileged information during demonstration collection, our system constrains the user to observing what the robot sees from its onboard cameras, resulting in more realistic trajectories and demonstrations that require minimal assumptions about the task at hand.

Second, recognizing that human data will be insufficient to cover all relevant states, we propose a new IL method that augments a trained policy with a learned error detector that can distinguish between in- and out-of-distribution states. When an agent encounters states previously unseen during training, our error detector can detect fatal errors and immediately stop the execution. In this way, our error detector can constrain the policy to execute only during states similar to what it has seen before, and prevent potentially unstable behavior from occurring during previously unseen states.

We demonstrate the potential of MOMART for mobile manipulation by generating the most general and largest dataset of mobile manipulation demonstrations publicly available: over 1200 demonstrations on five long-horizon tasks spanning expert, suboptimal, and few-shot generalization trajectories, totalling over 11 hours of simulation data. We use the dataset to train imitation learning policies that can achieve success rates greater than 45% and out-of-distribution detectors that achieve over 85% error detection success rate across all tasks.

In summary, our core contributions are as follows:

- We present MOMART, a novel teleoperation system that enables intuitive and expressive teleoperation of mobile manipulation robots,
- We collect a first-of-its kind continuous control dataset in a realistic simulated kitchen domain consisting of over 1200 successful demonstrations across five long-horizon tasks with multi-sensor modalities and ablation subsets with domain randomization,
- We train performant IL task policies that reach over 45% success across all tasks, and augment these policies with a learned error detector model that can accurately detect when the agent is in a failure state and immediately terminate, achieving over 85% precision and recall.

## 2  Related Work

**Robotic Teleoperation for Mobile Manipulation:** IL leverages human demonstrations to learn tasks such as stationary manipulation [10, 11, 12, 13] and navigation [14, 15, 16, 17]. Having human operators remotely control an agent, or teleoperation, is a common approach for collecting demonstrations. Teleoperation for MM is not easy to implement, as it requires enabling the user to control both navigation and manipulation, possibly simultaneously. Previous work on this

problem has explored using online click-through interfaces [18, 19, 20], muscle signals [21, 22], joysticks [23, 24, 25], tablets [26], and virtual reality interfaces [27, 28, 29, 30, 31, 32, 33, 34]. These approaches are either scalable or easy to use: web-based tools are widely available but are not well suited to demonstrate dexterous continuous control, while VR and other interfaces are intuitive to use but are not widely available. The new system we propose (Sec. 3) is both easy to use and scalable: the interface only requires a web browser for viewing and a phone that combines joystick and motion tracking capabilities for remote control of the agent [35].

**Autonomous Mobile Manipulation:** Algorithms for autonomous MM have been studied for decades [36, 37], and have primarily been addressed with either control [38, 39, 40, 41] or Task and motion planning (TaMP) [3, 42, 43, 44] approaches. Both are are able to generalize across different robots, environments, and robots, but control approaches are generally limited to short horizon tasks, and TaMP approaches depend on human specification of the symbolic actions and full information about the 3D structure of the environment. To address these limitations, learning-based approaches have recently been applied to MM to learn a direct mapping from raw sensory observations to actions [45, 46, 47]. Due to the lack of explicit planning, these works suffer in long horizon tasks involving heterogeneous types of actions such as grasping and moving objects.

Most related to our work, several works have leveraged IL for MM tasks [5, 6, 7, 8]. [5] presented a web-based tool for crowdsourcing a large scale dataset of MM tasks, and used it in combination with motion planning for execution on the robot. [6] and [7] collected RGBD observations of humans performing tasks such as door opening and tabletop object manipulation, and used hypergraph optimization and a search procedure respectively to adapt these trajectories to be executable by a robot. Lastly, [8] collected VR demonstrations of pick and place actions and extracted a sequence of symbolic actions and action parametrizations to adopt them for use on a robot. While these works demonstrate using IL for complex MM tasks, their approaches are limited to using the demonstrations to parametrize execution of action primitives, and so do not apply to the problem of learning a general visuo-motor policy of arbitrary manipulation actions as we do in this paper.

**Error Detection:** A common approach to prevent robotic agents from reaching states that harm the environment or themselves is to detect out-of-distribution (OOD) inputs to the agent [48]. Most relevant to our work, various approaches have been proposed for detecting OOD states using deep neural network-based architectures, such as direct training of an error prediction mode [49], uncertainty estimation for the policy [50, 50, 51, 52], or computing a reconstruction error for the input [53, 54, 55, 56]. In this work we follow the latter approach by training a conditional autoencoder to predict a future goal state given a current state and using its reconstruction error to determine if the robot is in a bad state. While prior works have similarly utilized such errors for collision avoidance, we go beyond them with a multi-modal prior that captures better the multiple solutions that humans demonstrated to the same goal, and apply the concept for the first time in a MM setup.

## 3  Mobile Manipulation RoboTurk (MOMART)

In this section, we present our approach for collecting demonstrations for MM. We first discuss RoboTurk [35], the precursor to our system, and then present our novel MOMART teleoperation system enabling remote and intuitive control of mobile manipulators.

**RoboTurk Overview:** RoboTurk [35, 57] is a platform that enables remote teleoperation of real or simulated robot arms. An operator connects to a server, receives live video stream with observations from the robot camera on their web-browser, and controls the robot's end-effector by moving a smartphone. The motion of the phone in Cartesian space (6 DoF, position and orientation) is tracked and mapped directly to the robot's end-effector motion, and leverages Web Real-Time Communication (WebRTC) to enable real-time control. RoboTurk has been used to collect large datasets on simulated [35] and real arms [57], accelerating IL for robot manipulation research [58, 59]. However, the platform has been limited to stationary arm manipulation.

### 3.1  MOMART

Due to the limitations of the original RoboTurk platform, we design MOMART, an extension ofthe original RoboTurk platform that enables intuitive control all the degrees of freedom of a mobile manipulator. In MOMART, we assume that the mobile manipulator consists of a single arm, a non-

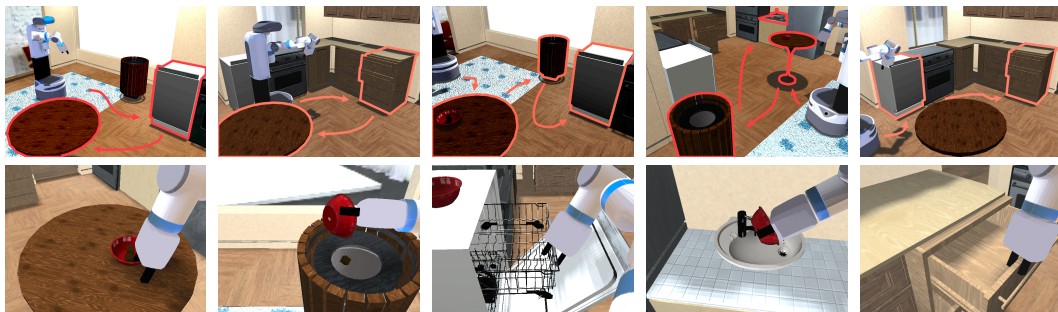

Figure 2: **Simulated Kitchen Tasks.** We present 5 challenging long-horizon mobile manipulation tasks requiring varied interactions with multiple household objects. *Top:* Tasks from left to right: Set Table from Dishwasher, Set Table from Dresser, Table Cleanup to Dishwasher, Table Cleanup to Sink, Unload Dishwasher to Dresser. *Bottom:* Our tasks evaluate an agent's ability to execute diverse sets of manipulation and navigation skills, including accurate arm positioning and contact-rich, arm-base coordinated interaction of large constrained mechanisms. Teleoperators are limited to this ego-centric view, highlighting partial observability.

holonomic base, and a controllable head, such as the Fetch or PAL Tiago robots. We model our system close to real-robot hardware and plan to extend it to the real-world post-pandemic (Sec. A.10).

As evidenced by prior MM teleoperation platforms, designing a well-balanced interface can be challenging for multiple reasons: fully exploring MM requires simultaneous control of base and arm motion, but allowing control of all possible degrees of freedom can overload the operator and degrade the quality of the resulting demonstrations. Moreover, unlike static manipulation with a global fixed frame of reference, mobile manipulators often only have access to a local frame of reference, modified only by controlling the head. Finally, the long-horizon nature of MM increases the likelihood of demonstrations diverging over time, resulting in sparsely distributed end states.

We showcase our smartphone interface design in Fig. A.1 with all of these challenges in mind. Our platform is easy to use, enabling simultaneous 6DOF arm end-effector control through smartphone motion and base locomotion control through the on-screen joystick. To address the head control problem, we fix the head pan (horizontal) joint and allow the tilt (vertical) joint to automatically maintain the end-effector in the central area of the head camera frame. Lastly, recognizing that teleoperated robots can fall into bad configurations, we include a button-triggered arm-reset that generates a trajectory to move the arm to a pre-defined stable initial joint configuration. The reset helps regularize the arm state distribution observed during teleoperation and increases the consistency between demonstrations, and is leveraged in our final IL learning system with error detection to try and recover from mistakes (Sec. 4.3). Altogether, MOMART enables collecting of teleoperated demonstrations from remote locations with an intuitive interface that allows to simultaneous navigation and manipulation in complex environments. Further details can be found in Appendix A.4, and we also conduct a user study to qualitatively evaluate our system in Appendix A.5.

## 3.2 Simulated Kitchen Dataset

To evaluate the teleoperation capabilities of MOMART for MM and generate data for our novel IL with error detection algorithm (Sec. 4), we create five realistic multi-stage simulated household kitchen tasks and collect a large-scale multi-user demonstration dataset. We design the simulated tasks in a realistic kitchen environment using PyBullet [60] and the iGibson [61, 62] framework with a Fetch [63] robot that must manipulate a bowl. Across all tasks, the robot's initial pose and bowl location is randomized between episodes. An overview of our tasks can be seen in Fig. 2 and are summarized in the following:

**Table Cleanup to Dishwasher / Sink:** The robot must navigate to the table, pick up the bowl with trash in it, navigate to the trashcan, and empty the trash. Then, it must either (a) take the bowl to the dishwasher, open the dishwasher and pull out the tray, and place the bowl into the tray, or (b) take the bowl to the sink and drop it in the basin. In addition to the robot and bowl, the trash can's pose is also randomized between episodes. These tasks evaluate an agent's ability to execute a diverse range of manipulation and navigation skills, including accurate arm positioning and contact-rich, arm-base coordinated interaction of large constrained mechanisms.

**Table Setup from Dresser / Dishwasher:** The robot must first either (a) navigate to the dresser, and search for the bowl by opening each drawer, or (b) navigate to and open the dishwasher, and then pull out the tray. Afterwards, in both cases, it must grab the bowl, and navigate to the table and drop off the bowl. In addition to local randomization, the bowl's location is also randomized across the drawers between episodes. Searching the dresser evaluates an agent's ability to contextualize its observations on prior actions.

**Unload Dishwasher to Dresser:** The robot must first navigate to the dishwasher and grab the bowl. Then, it must navigate to the dresser, open the top drawer, and place the bowl inside. This task evaluates an agent's ability to avoid obstacles based on estimated visual states.

For each task, we use MOMART to collect over 110 successful demonstrations per task from both an expert and suboptimal operator, as well as additional few-shot generalization subsets consisting of over 20 demonstrations where key task furniture items have changed significantly their locations, producing a dataset that includes over **1200 successful demonstrations** totalling **over 11 hours of data**. This is a first of its kind large-scale human demonstration dataset of continuous control collected on realistic long-horizon tasks in the MM setting, and we hope this dataset facilitates future research in IL for MM.

# 4 Learning Mobile Manipulation from Human Demonstrations

In this section, we present our approach for IL for MM. After some preliminaries, we introduce our modified temporal network architecture with more efficient temporal abstraction for IL of MM, and then propose an error detection model that can distinguish between in- and out-of-distribution states to alleviate the challenges of the unbounded state space in MM.

## 4.1 Preliminaries

**Partial Observability**. We formalize the problem of solving a robot MM task as an infinite-horizon discrete-time Partially Observable Markov Decision Process (POMDP). At every step, an agent in state $s_t$ receives an observation $o_t$ and uses a policy $\pi$ to choose an action, $a_t = \pi(o_t, o_{t-1}, ...)$, which moves it to state $s_{t+1}$ according to the state transition distribution.

**Imitation Learning:** We train a visuo-motor continuous control policy for MM with a variant of Behavioral Cloning (BC) [64]. The policy maps observations to base and arm actions for the mobile manipulator. BC trains a policy, $\pi_\theta(o)$, from a set of demonstrations, $\mathcal{D}$, by minimizing the objective: $\arg\min_\theta \mathbb{E}_{(o,a)\sim\mathcal{D}}||\pi_\theta(o) - a||^2$. We base our policy on a variant of BC that leverages temporal abstraction, BC-RNN, in which the policy is parameterized by a recurrent neural network (RNN) that is trained on $T$-length temporal observation-action sequences to produce an action sequence, $a_t, \ldots, a_{t+T-1}$. To account for the possibility of multi-modal possibilities in a given state, actions are parameterized by a Gaussian Mixture Model (GMM) distribution [65], $a_t \sim \sum_i^N w_i \mathcal{N}(\mu_i, \sigma_i)$, where actions are sampled from amongst $N$ weighted individual Gaussian distributions.

## 4.2 TieredRNN

Prior work on learning from human demonstrations in robotic manipulation domains has shown substantial benefits from models leveraging temporal abstraction [58, 66]. Inspired by these works, we extend the RNN model from BC-RNN into a multi-layered variant (TieredRNN). This new variant integrates multiple layers operating at varying timesteps to better streamline information flow from timesteps early on in a given sequence to timesteps much further downstream, which can be useful for our long-horizon MM tasks.

The TieredRNN consists of $N$ individual RNN layers ("tiers"), with corresponding timestep periods $\tau_1, ...\tau_N, \quad \tau_1 < ... < \tau_N$. For given sequence of length $T$, $t = t, t+1, ..., t+T$ and corresponding states $s_t, s_{t+1}, ..., s_{t+T}$, layer $i$ updates its hidden state if $\tau_i \mod t = 0$ and outputs a $M$-dim encoding vector $z_i$, which gets passed in addition to $s_t$ to the immediate proceeding $i-1$ layer. The output of the final layer $i = 1$ is the overall output of the TieredRNN. In this way, information at varying levels of temporal abstraction can be preserved across many timesteps and better inform an agent of past context. We refer to BC agents using this temporal structure as BC-TieredRNN agents.

| Task | ReLMoGen [1] | BC-RNN Suboptimal | BC-TieredRNN Suboptimal | BC-RNN Expert | BC-TieredRNN Expert |
|------|------------|-------------------|-------------------------|---------------|---------------------|
| Table Cleanup to Dishwasher | $0.0 \pm 0.0$ | $16.0 \pm 5.5$ | $20.8 \pm 5.0$ | $44.4 \pm 4.0$ | $\mathbf{47.8 \pm 7.3}$ |
| Table Cleanup to Sink | $0.0 \pm 0.0$ | $10.4 \pm 5.2$ | $8.9 \pm 2.9$ | $54.4 \pm 7.1$ | $\mathbf{61.1 \pm 3.1}$ |
| Table Setup from Dresser | $0.0 \pm 0.0$ | $20.0 \pm 0.9$ | $17.8 \pm 4.8$ | $64.8 \pm 5.8$ | $\mathbf{68.1 \pm 3.4}$ |
| Table Setup from Dishwasher | $0.0 \pm 0.0$ | $63.7 \pm 7.7$ | $61.1 \pm 3.1$ | $\mathbf{68.5 \pm 7.6}$ | $66.2 \pm 2.3$ |
| Unload Dishwasher to Dresser | $0.0 \pm 0.0$ | $11.9 \pm 4.5$ | $31.1 \pm 7.4$ | $\mathbf{56.0 \pm 7.0}$ | $47.8 \pm 8.1$ |

[1]We include more extensive results in Appendix A.6

Table 1: **Simulated Kitchen Tasks Results:** IL can solve all of our multi-stage tasks, whereas RL (ReLMo-Gen) cannot solve any task and in all tasks is unable to even grasp the bowl as is detailed in Appendix A.6. When trained on expert data, our BC-TieredRNN model can achieve over 45% success rate over all tasks, and outperforms all other baselines on a majority of the tasks.

### 4.3 Error Detection and Intervention Functionality

One of the central challenges of IL for MM is the unbounded state space that the robot can explore, since states that sufficiently differ from collected human demonstrations can cause an IL policy to quickly degrade. This common covariate shift problem [9] of IL is thus exacerbated by the MM setup. While it is difficult to alleviate this problem without collecting additional online samples, we propose a simple but effective method for detecting errors and improving the overall safety of policy execution by leveraging our demonstration data as a strong prior for distinguishing between in- and out-of-distribution states (Fig. 1).

**Detecting Errors.** Similar to prior work, our error detector $\Phi$ leverages a variational autoencoder (VAE) [67] and its reconstruction error $\epsilon$ as a proxy for distinguishing in- and out-of-distribution states. In our setting, this means that with sufficient training data and capacity, $\Phi$ should learn to reconstruct similar states to those observed from the demonstrations with low $\epsilon$. We leverage $\epsilon$ for distinguishing between in- and out-of-distribution states: abnormally high $\epsilon$ likely correspond to unseen states and can be interpreted as failure modes during rollouts. Implementation details can be found in Appendix A.2.

**Intervening During Errors.** We implement two discrete intervention actions that can be triggered when an error is detected. `recover` is triggered the first time an error is encountered, and moves the agent to a default pose to allow it to re-attempt execution. If the `recover` action fails to bring the error level below the threshold, or if $K$ errors have been detected within a given rollout, the error detector executes `terminate`, which immediately terminates the episode. In this way, even if an error is unrecoverabale, the agent can detect and respond to its circumstance (for example, asking a human for help). Crucially, because our error detector and policy are not provided any additional online data, we do not expect the `recover` action to consistently succeed. However, we do expect our `terminate` intervention to be consistent and reliable as it relies solely on our error detector's ability to consistently detect out-of-distribution states; this is essential for mitigating potentially harmful policy behavior resulting from covariate shift. Algorithm 1 formalizes our method.

## 5 Experimental Evaluation

In this section, we first evaluate our IL algorithm for MM, BC-TieredRNN, and compare to baselines of IL and RL, BC-RNN, and ReLMoGen [45]. ReLMoGen is a state-of-the-art RL-based algorithm that leverages discrete actions with a motion planner. Then, we train and evaluate our error detector and recovery for MM, and show that it can accurately detect out-of-distribution states and either help the agent recover to in-distribution states or know when to terminate due to unrecoverable errors. Videos and other results can be seen at https://sites.google.com/view/il-for-mm/home. Specific training and hyperparameter details can be found in Appendix, Sec. A.1 and A.3.

### 5.1 Simulated Results: Solving Long-Horizon Mobile Manipulation Tasks

For each task, we train each model for 30 epochs and record the average top three best evaluation success rates from 30 episodes aggregated over 3 seeds. For ReLMoGen, we use a shaped reward for each task and train the agent for 200K time steps. Our results can be seen in Table 1.

**Imitation Learning Outperforms Reinforcement Learning.** We observe that ReLMoGen cannot solve any task despite its lifted action space and shaped reward. This highlights the exploration burden exacerbated in the MM setup, which can cause current state-of-the-art RL methods to completely flatline. In contrast, our IL methods can perform well and achieve success learning end-to-end visuo-motor policies on these multi-stage MM tasks.

On the expert dataset, we find that our BC-TieredRNN model outperforms the BC-RNN baseline in the tasks where memory is critical: *Table Cleanup to Dishwasher* and *Table Cleanup to Sink*, both of which have randomized locations of the trash bin, and *Table Setup from Dresser*, where the bowl can be located in multiple drawers. In these tasks where the agent must remember where it has been in the past, we hypothesize that the TieredRNN is better equipped to allow long-term information flow through its skip-connection architecture. In contrast, in the tasks that do not require randomization (*Table Setup from Dishwasher* and *Unload Dishwasher to Dresser*), the TieredRNN provides no benefit and is instead outperformed by BC-RNN. We further contextualize our usage of the TieredRNN model in Appendix A.7.

**Demonstration Quality Dictates Success.** With the exception of *Table Setup from Dishwasher*, we observe on average over 200% increase in success rate when training on the same number of expert demonstrations compared to suboptimal demonstrations. In the latter case, the operators were new to using the MoMART system, whereas the expert operator had prior experience. This both showcases the ability of our platform to produce impressive results on multiple long-horizon MM tasks given experience, and also highlights the importance of the teleoperation interface for collecting high-quality demonstrations that better facilitate learning.

## 5.2 Simulated Results: Detecting Errors During Rollout

For each task, we train our error detector and set a uniform error threshold $\psi = 0.05$ based on analysis from a small number of expert rollouts. Because the BC-TieredRNN model performed the best on the majority of tasks, we utilize this model for error detection evaluation. We evaluate our error detector's performance when paired with the trained policy model, and record the precision ($\frac{n_{pos_t}}{n_{pos_t} + n_{pos_f}}$) and recall ($\frac{n_{pos_t}}{n_{pos_t} + n_{neg_f}}$). Because we are in simulation, we can deterministically evaluate the counterfactual between the model augmented with the error detector and the same model without. A true / false positive occurs when the error detector detects an error when the original model fails / succeeds, respectively. Likewise, a true / false negative occurs when the error detector does not trigger during a successful /failed rollout, respectively. Metrics are aggregated over the 3 policy seeds each evaluated on 30 rollouts with mean and standard deviations shown in Table 2.

**Errors are Reliably and Robustly Detected.** We find that our error detectors can consistently detect errors and achieve over 85% precision and recall rates across all tasks when trained on the expert dataset. The high precision means that our error detectors can accurately distinguish between error and non-error states, and the high recall means that it can also reliably detect true errors when they occur. This is especially important in MM domains where learned agents can easily diverge into unseen states during rollouts and become erratic if left unchecked.

While the error detectors occasionally misfire, and causes a marginal decrease in success, we observe that the success rates remains generally consistent. This highlights that our error detector is not overly conservative and does not prematurely terminate episodes when there is no true error at hand. Interestingly, in some cases, such as in *Table Setup from Dresser*, the `recover` action of our error detector is able to result in marginal policy improvement. While unexpected, this suggests that similar methods bringing a robot back into well-known states may provide benefits for MM agents.

**Data Diversity Impacts Error Detection** When trained on suboptimal data, our error detectors surprisingly achieve over 90% precision across all tasks. However, the recall rate substantially suffers. We postulate that this is due to the noisy nature of the suboptimal data encompassing a wider distribution of states, augmenting the resulting error detectors' abilities to reject non-error states while making it more difficult to consistently detect true error states.

**Our Error Detector is Interpretable and Easily Tuned.** While we deployed all error detectors with the same uniform error threshold value $\epsilon$, each error detector can be individually tuned in an interpretable way. For example, lowering $\epsilon$ increases the error detector's sensitivity to observation irregularities, thereby improving the likelihood of detecting true errors (increased recall), at the potential cost of additional spurious error triggers (decreased precision). This can be useful in

| Dataset | Task | SR (no ED) | SR (ED) | ED Precision | ED Recall |
|---|---|---|---|---|---|
| Expert | Table Cleanup to Dishwasher | $34.4 \pm 10.3$ | $32.2 \pm 8.7$ | $96.4 \pm 2.6$ | $100.0 \pm 0.0$ |
| | Table Cleanup to Sink | $51.1 \pm 4.2$ | $51.1 \pm 4.2$ | $97.2 \pm 3.9$ | $86.5 \pm 10.3$ |
| | Table Setup from Dresser | $63.3 \pm 7.2$ | $65.6 \pm 8.7$ | $100.0 \pm 0.0$ | $85.9 \pm 10.0$ |
| | Table Setup from Dishwasher | $51.1 \pm 6.8$ | $47.8 \pm 11.0$ | $88.2 \pm 10.2$ | $100.0 \pm 0.0$ |
| | Unload Dishwasher to Dresser | $36.7 \pm 10.9$ | $33.3 \pm 9.8$ | $85.8 \pm 5.3$ | $100.0 \pm 0.0$ |
| Suboptimal | Table Cleanup to Dishwasher | $16.7 \pm 4.7$ | $22.2 \pm 8.3$ | $93.4 \pm 5.3$ | $76.9 \pm 18.8$ |
| | Table Cleanup to Sink | $1.1 \pm 1.5$ | $1.1 \pm 1.5$ | $98.4 \pm 2.2$ | $66.3 \pm 7.1$ |
| | Table Setup from Dresser | $7.7 \pm 6.3$ | $15.6 \pm 4.2$ | $97.6 \pm 3.4$ | $58.7 \pm 7.4$ |
| | Table Setup from Dishwasher | $52.2 \pm 4.2$ | $48.9 \pm 5.7$ | $90.0 \pm 7.7$ | $95.2 \pm 6.7$ |
| | Unload Dishwasher to Dresser | $26.7 \pm 2.7$ | $26.7 \pm 2.7$ | $94.6 \pm 3.9$ | $44.2 \pm 14.1$ |
| Few-Shot Generalize | Table Cleanup to Dishwasher | $13.3 \pm 4.7$ | $6.7 \pm 2.7$ | $91.2 \pm 4.9$ | $94.8 \pm 3.7$ |
| | Table Cleanup to Sink | $24.4 \pm 6.8$ | $11.1 \pm 6.8$ | $80.6 \pm 3.7$ | $98.7 \pm 1.9$ |
| | Table Setup from Dresser | $8.8 \pm 8.3$ | $7.8 \pm 8.7$ | $98.0 \pm 2.8$ | $64.4 \pm 6.7$ |
| | Table Setup from Dishwasher | $40.0 \pm 16.3$ | $24.4 \pm 12.9$ | $78.2 \pm 10.6$ | $100.0 \pm 0.0$ |
| | Unload Dishwasher to Dresser | $2.2 \pm 1.6$ | $2.2 \pm 1.6$ | $98.9 \pm 1.6$ | $100.0 \pm 0.0$ |

Table 2: **Error Detection Results** When trained on expert data, our error detectors can accurately detect errors, achieving over 85% precision and recall across all tasks. With suboptimal data, the error detectors can similarly reject non-error states consistently but can struggle to detect true error states. Lastly, the error detectors can adapt to the few-shot generalization setting, maintaining high precision and recall despite the generalized policy's significant deterioration.

situations where safety is critical and the additional confidence in reliable error detection is worth the cost of lowered success rate. Further results comparing our method against potential alternatives can be seen in Appendix A.8.

## 5.3 Simulated Results: Few-Shot Generalization

Lastly, we evaluate the generalizability of our policy and error detector by taking the models trained on the expert data and finetuning them on the few-shot demonstrations exhibiting major distribution shift, where the critical furniture objects (dishwasher, dresser, and sink) have swapped places. Similar to Sec.5.2, we evaluate the success with and without the error detector, and also report the corresponding detection metrics, shown at the bottom of Table 2.

**Imitation Learning Has Potential to Generalize With Little Additional Data.** While the success rates drop significantly compared to the original baselines, we find that our model can still solve the task given only a limited number of successful demonstrations in this generalized setting. These results show promise for IL to lower the data burden for learning general visuomotor policies in similar MM settings as our kitchen environment, where the discrete number of unique object instances is finite but the combinatorial possibilities are huge. Appendix A.9 highlights the benefits of finetuning versus training solely on the few-shot demonstrations.

**Error Detection Can Adapt to Distribution Shifts.** Even in this generalized setting, our error detector still performs well, and often achieves more than 80% precision and 90% recall across all tasks. This shows how our detector, once initially trained, can quickly adapt its modeled state distribution to account for new training data and still accurately detect true error states while ignoring non-error states.

## 6 Conclusion

We presented three contributions for IL in MM setups. First, we introduced MOMART, a novel teleoperation platform for MM. Second, we used MOMART collect a first of its kind large-scale MM dataset of continuous control. And third, we presented variants of IL and error detection for the MM setup that train performant visuo-motor policies and accurately detect errors. Our next step after the pandemic is to bring our new teleoperation interface to control a real mobile manipulator (Sec. A.10). We hope our diverse dataset can provide researchers the accessible means to investigate many other important problems in MM from an IL perspective.

**Acknowledgments**

We would like to thank Jiangshan Li for her help in setting up the initial RoboTurk prototype.

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
