# OpenReview forum: "Error-Aware Imitation Learning from Teleoperation Data for Mobile Manipulation"
_robot-learning.org/CoRL/2021/Conference — CoRL2021 Poster_

### Official Review · Reviewer_KoyB · 2021-07-13

**Originality:** Fair
**Technical Quality:** Good
**Clarity Of Presentation:** Very Good
**Impact:** 3

**Recommendation:**

Weak Accept: I recommend accepting the paper, but will not argue for my recommendation if the majority of other reviewers have a different opinion.

**Summary:**

The paper addresses learning mobile manipulation tasks by 1) building a smartphone teleoperation interface that enables human operators to demonstrate mobile manipulation tasks in simulation, 2) training a visuomotor policy on such collected demonstrations, and 3) training an error-detection classifier to detect when the learned policy had incurred an unseen state and either attempts to recover or terminates task execution. Experiments were done in a simulated kitchen domain, where a fetch robot was tasked to transport objects by interacting with a variety of articulated objects. A total of 1200 demonstrations were collected, and the learned policies achieved task success rate of 45%.

**Issues:**

Please see Strength and Weaknesses for revision suggestions. The website link was not operational at the time of review.

**Reviewer Expertise:**

Good: General knowledge of the area

**Strengths And Weaknesses:**

Strengths:
-	The paper’s motivation is well founded – learning long-horizon mobile manipulation tasks could benefit from more robust human priors.
-	The fact that the demonstrator is not using privileged information is a plus – both the demonstrator and the robot is given the same single-camera view.
-	An easy to use interface for providing mobile manipulation demonstration data is clearly beneficial for the field.
-	The paper is clearly written.

Weaknesses:
-	My main criticism is the paper’s seeming lack of focus. While it builds a system that can collect mobile manipulation demonstrations and train imitation learning policies, it does not study these components in depth.
-	On the teleoperation interface, a more thorough user study would have made the paper more convincing. There are many questions such a study can help answer. What is special about giving demonstrations for mobile manipulation as compared to stationary demonstration? Were task strategies pre-determined or did each demonstrator come up with their own? How varied were the demonstrations, and how noisy were the collected trajectories? Were any data cleaning and filtering done to preprocess demonstration data? What’s the success rate of human demonstrators? Could the camera be moved independently of the robot? How many trials did it take for a new human operator to achieve task success?
-	On imitation learning, while the use of RNN seems appropriate given the partial observability aspect of single-view camera inputs, it would still be helpful to run a baseline comparison against a non-RNN version with a window of observations as input. The claim that the Tiered-RNN architecture performs better than the generic RNN architecture is also not obvious, at least not from Table 2.
-	On covariate shift - the use of error detection to address covariate shift is not entirely satisfying. There are many well established and simple-to-implement methods for reducing covariate shift, from the DAgger algorithm to noise injection during demonstrations, and it is unclear why the authors did not experiment with them. One potential justification may be that the paper is not completely focused on robust imitation learning, but that further highlights the lack of focus in the paper


**Summary Of Recommendation:**

I recommend weak reject. In my opinion, the only novel part about the paper is its mobile manipulation teleoperation interface (and perhaps the 1200 collected demonstrations), but the paper does not perform thorough experiments and analyses of this interface. The results from the trained manipulation policies, while interesting, do not bring a new understanding of the problem.

**Update** After the author's response with additional clarifications and the new user study, which helped to address my original concerns and better contextualize the paper, I am changing my recommendation to a weak accept.

---

> ### Author Response · Authors · 2021-08-28
> **Response to Reviewer KoyB**
>
> Please note that our response has been split into three parts due to space constraints. (Part 1/3)
>
> Thank you for the insightful feedback and questions! We appreciate your patience in awaiting a response; we’ve conducted multiple follow-up experiments in response to feedback, and it has taken time to recruit people and conduct a user study on our MoMaRT system.
>
> We have updated our manuscript and corresponding supplementary materials (appendices); all modified / added words are highlighted in red for easy viewing.
>
> We are glad to hear that you think our interface and dataset can be useful for the robotics field. We’d also like to apologize for our website link being down -- we were unaware that there were access restrictions placed when we originally published. We encourage you to (re-)visit our website (https://sites.google.com/view/il-for-mm/home) to see additional helpful supplemental material.
>
> Below, we have tried to thoroughly respond to each of your questions and concerns. Please let us know if any information is lacking or unclear:
>
> - “My main criticism is the paper’s seeming lack of focus. While it builds a system that can collect mobile manipulation demonstrations and train imitation learning policies, it does not study these components in depth.”
>
> This is a good point, and something we acknowledge. While we address the specific issues raised in the following sections, we’d like to provide perspective on our approach as a whole. In general, imitation learning for mobile manipulation has not been extensively tackled in any setting, and we found little prior work that addresses this problem with minimal sets of assumptions. Thus, our intention is to build a fully end-to-end imitation learning for mobile manipulation pipeline and evaluate this system overall to study its effectiveness, rather than focusing on any single component. We hope to provide in a three-fold way: by (a) providing a flexible method for collecting mobile manipulation demonstrations in a realistic setting (i.e.: no privileged information or magic actions), (b) publishing a novel large dataset specifically constructed to offer multiple avenues for future works (e.g.: compositionality, transfer learning), and (c) a set of baseline methods that both validate imitation learning’s tractability in mobile manipulation and potential for mobile manipulation’s key challenge (large state space / covariate shift) to be addressed. Our goal is to minimize the assumptions of our problem setup so that our work can be widely applicable and easy to build upon.
>
> - “On the teleoperation interface, a more thorough user study would have made the paper more convincing. There are many questions such a study can help answer. What is special about giving demonstrations for mobile manipulation as compared to stationary demonstration? Were task strategies pre-determined or did each demonstrator come up with their own? How varied were the demonstrations, and how noisy were the collected trajectories? Were any data cleaning and filtering done to preprocess demonstration data? What’s the success rate of human demonstrators? Could the camera be moved independently of the robot? How many trials did it take for a new human operator to achieve task success?”
>
> This is very true. While there is pre-existing work that has already conducted user studies on the original RoboTurk interface which our platform is built upon (https://arxiv.org/pdf/1911.04052.pdf) that showcases its efficacy, we conducted a small-scale user study with 10 new users to provide qualitative and quantitative evaluation of our platform. In general, users were usually able to feel comfortable with the interface within 15 minutes, but struggled to confidently teleoperate the mobile manipulator, though users were only allotted 30 minutes of playtesting time. Much of the user feedback highlighted the difficulties associated with the POMDP mobile manipulation setting, such as limited field of view and difficulty executing precise arm trajectories to operate articulated objects. All users found the arm reset functionality to be beneficial during teleoperation, validating this key design decision in our interface. Despite the challenges faced with our system and difficult domain, the merits are clear in the ultimate scale of data and learning results we have already been able to produce, and hope that future work can continue to build upon our platform to increase its efficacy for users. We encourage you to see the full details and discussion of results of our user study in Appendix A.5.

---

> > ### Author Response · Authors · 2021-08-28
> > **Response (Part 2)**
> >
> > Please note that our response has been split into three parts due to space constraints. (Part 2/3)
> >
> > With regards to the specific questions:
> >
> > > “What is special about giving demonstrations for mobile manipulation as compared to stationary demonstration?”
> >
> > We highlight two key details that distinguish static (stationary) manipulation demonstrations from mobile manipulation demos:
> >
> > (a) Larger, multi-modal action space. In mobile manipulation, agents are able to actuate their locomotive (base) and manipulative (arm) components simultaneously. As humans, we do this naturally, but it has been surprisingly nontrivial to apply this to collecting actual mobile manipulation demonstrations (https://ieeexplore.ieee.org/document/7139971, http://ais.informatik.uni-freiburg.de/publications/papers/welschehold17iros.pdf, http://ras.papercept.net/images/temp/IROS/files/2587.pdf ).
> >
> > (b) Dynamic workspace. Critically, either the action or observation frame of reference no longer becomes fixed. Either an external, global camera is used, which can make teleoperation manipulation difficult since as the robot rotates, the same teleoperated movement would result in different actions; or, conversely, an onboard robot head camera is used (as in our case), which results in moving observation frames of reference that can additionally be rotated by head joints.
> >
> > Together, this makes the transition from static manipulation to mobile manipulation demonstrations both nontrivial and increasingly complex.
> >
> > > “Were task strategies pre-determined or did each demonstrator come up with their own?”
> >
> > Task strategies were determined by each demonstrator; however, demonstrators also modified their strategy if they found a more optimal solution (e.g: for opening the dishwasher, most operators found that opening from the side was significantly easier than opening from the front). An example of this can be seen on our website.
> >
> > > “How varied were the demonstrations, and how noisy were the collected trajectories?”
> >
> > Environment initializations were randomized with uniform noise and generally provided the most variation, as high-level demonstrator strategies usually became consistent once converged. Noise varied between demonstrator, with expert demonstrations having much fewer grasps and mis-coordinated motions compared to the suboptimal demonstrators.
> >
> > > “Were any data cleaning and filtering done to preprocess demonstration data?”
> >
> > No data cleaning or postprocessing (e.g.: trajectory smoothing) was done, other than discarding the failed demonstration episodes.
> >
> > > “What’s the success rate of human demonstrators?”
> >
> > While we do not have a success rate for our main dataset, we found in our user study that new users who were able to solve the Table Setup from Dishwasher within the allotted time averaged 56% success rate. However, we highlight that most users only attempted to solve the task for around 15 minutes, and that this task is quite difficult due to it being multi-stage and requiring precise movements and large-rotational trajectories. Thus, we expect these users to show tangible improvements in task success rate if given more playtesting time.
> >
> > > “Could the camera be moved independently of the robot?”
> >
> > We could have utilized an external camera used for teleoperation, but chose to constrain the teleoperator to seeing what the robot sees from its head camera in order to enforce the POMDP setting and prevent the user from leveraging privileged global information that the robot cannot access.
> >
> > > “How many trials did it take for a new human operator to achieve task success?”
> >
> > 50% of users were unable to achieve a task success within the allotted time. Of the remaining 50% users that were able to solve the task, all but one solved it during the first trial. The remaining person solved it on the second trial. However, of the people who were unable to achieve a task success, most of them were able to at least partially solve the task, and would have likely been able to solve the full task if given additional playtest time.
> >
> > - “On imitation learning, while the use of RNN seems appropriate given the partial observability aspect of single-view camera inputs, it would still be helpful to run a baseline comparison against a non-RNN version with a window of observations as input. The claim that the Tiered-RNN architecture performs better than the generic RNN architecture is also not obvious, at least not from Table 2.”

---

> > > ### Author Response · Authors · 2021-08-28
> > > **Response (Part 3)**
> > >
> > > Please note that our response has been split into three parts due to space constraints. (Part 3/3)
> > >
> > > Indeed, Table 2 shows that the TieredRNN model only performs marginally better than the RNN baseline. To clarify, we do not claim that the TieredRNN model is generally a better option than RNN in the paper. However, we choose to use the TieredRNN model because it provided marginal benefits in our specific setup. We would also like to provide some further context that might not be as readily apparent from our paper, and which we have now included in Appendix A.7. We found during preliminary hyperparameter sweeping that the TieredRNN scales better than the RNN model given similar total parameter counts; that is, further increasing the RNN model’s parameter count via the hidden dimension size generally resulted in policy degradation. Moreover, we found the TieredRNN model to also provide performance gains on previous iterations of tasks that were not included in this work. These empirical observations led us to deem the TieredRNN valuable enough to include as our final evaluation model, despite providing marginal benefits in this specific setting. We hope to continue iterating on this model in future work.
> > >
> > > With regards to the non-RNN model, we do not expect it to outperform our RNN models. During experiment prototyping, we ran a hyperparameter sweep over our RNN’s sequence length, and found that using a large length (in our case, 50) was crucial for achieving our policy results. Moreover, other prior work has found that using an RNN can show impressive performance in complex visual scenes (https://arxiv.org/pdf/1911.00357.pdf). Thus, a stacked frames method with access to a small number of past frames is unlikely to outperform our method. Alternatively, more complex methods such as convolutions over longer histories may be a more viable alternative, but this is outside the scope of this work.
> > >
> > > - “On covariate shift - the use of error detection to address covariate shift is not entirely satisfying. There are many well established and simple-to-implement methods for reducing covariate shift, from the DAgger algorithm to noise injection during demonstrations, and it is unclear why the authors did not experiment with them. One potential justification may be that the paper is not completely focused on robust imitation learning, but that further highlights the lack of focus in the paper”
> > >
> > > We first apologize for the lack of clarity in our paper, and in addition to refining the wording in our paper, would like to clarify the intended role of our error detector. We agree that there are many common and effective methods for explicitly improving online policy performance. However, these methods either may require additional online samples during evaluation (DAgger), or may require a greater number of demonstrations to begin with (noise injection). Human data is expensive, and this is especially compounded in mobile manipulation, where tasks are often long-horizon with wide state space coverage (as in our case). In contrast, our method requires no additional online samples, and is fully trained offline. We highlight that our method is not intended to consistently compensate for errors during intervention (reducing covariate shift), but rather detect and terminate execution to prevent potentially dangerous out-of-distribution policy behavior from occurring that could damage the robot and / or its surroundings (addressing covariate shift).
> > >
> > > However, we did evaluate our error detection mechanism against other fully-offline methods, to show that our method is in fact relatively more performant compared to obvious alternatives. In addition to our main reconstruction loss metric, we consider another loss metric (KL Divergence loss for the VAE model), latent metrics (VAE Encoder Variance Mean / Max values), and a policy uncertainty metric (policy action log probability). We find that our model strongly outperforms all of these methods, and is the only model able to achieve consistently high error detection precision and recall rates. This further validates prior work (https://arxiv.org/pdf/2012.00201.pdf, http://www.roboticsproceedings.org/rss13/p64.pdf, https://www.sciencedirect.com/science/article/pii/S016516841300515X ) that has found reconstruction error to be a viable metric for detecting errors, and also highlights the value of our method's ability to be easily tuned and robust across multiple tasks, which is a property not apparent in the other baselines. (Indeed, we found that for the latent and log probability methods, the error signal was very noisy and did not necessarily transfer well between tasks given the same threshold error value). Exact results and further analysis can be found in Appendix A.8.

---

> > > > ### Author Response · Authors · 2021-08-30
> > > > **Friendly Follow-Up**
> > > >
> > > > Dear reviewer KoyB, we apologize for posting so close to the deadline but we were working hard to obtain empirical results and add some of the experiments you recommended. We hope you had time to look at those and our replies. Did we address your concerns? Are there any other comments or clarifications that we could provide, that would make you consider increasing your score? Best regards, authors of Paper193

---

> > > > > ### Comment · Reviewer_KoyB · 2021-08-31
> > > > > **Rebuttal Response**
> > > > >
> > > > > I would like to thank the authors for the detailed explanations and performing the new user study. They helped to address the concerns in the original review and better contextualize the paper. I have updated the recommendation to a weak accept.

---

### Official Review · Reviewer_dB9j · 2021-07-21

**Originality:** Good
**Technical Quality:** Good
**Clarity Of Presentation:** Very Good
**Impact:** 4

**Recommendation:**

Weak Accept: I recommend accepting the paper, but will not argue for my recommendation if the majority of other reviewers have a different opinion.

**Summary:**

The paper presents an innovative imitation learning method with error detection in the context of mobile manipulation. It proposes a novel architecture, TieredRNN, which is a multi-layered variant of BC-RNN. It is able to perform significantly better than the current SOTA methods for a series of real-world mobile manipulation tasks. On top of that, it presents a way to detect out-of-distribution (OOD) inputs using VAE reconstruction error and perform intervening error corrections for the agents.

**Issues:**

Related to the Weaknesses section above:

- More detailed explanation and justification of presenting the TieredRNN methods

- More justified explanation on how error correction relates to success rate

On error correction strategies:

- The current interventions when an error is detected are hardcoded programming if I interpret it correctly - only "recover" and "terminate". I'm curious of whether the authors have considered more versatile intervention methods, e.g. human monitoring and tele-operated corrections?

**Reviewer Expertise:**

Good: General knowledge of the area

**Strengths And Weaknesses:**

Strengths:

- It is quite impressive that BC based methods perform much better than RL based baselines in a challenging domain like mobile manipulation. It shows the effectiveness of BC + RNN in long sequential tasks and gives room for future exploration into this line of work.

- Successfully evaluated the effect of suboptimal demonstration data

- Presented a novel error detection methods based on VAE construction error

Weaknesses:

- On choice of model architectures: from the experimental results, it is not entirely clear to me how TieredRNN perform better than BC-RNN, and it seems that BC-RNN (which I assume to be a simpler structure) should already give quite impressive results. Is there any reason that the authors choose the more complex model structure of TieredRNN over simplicity? It would be better if there could be more explanations in the paper.

- On experimental results: According to Table 2, the error detection results are quite impressive, but they do not help success rate that much. In fact in many sections, the success rate decreases because of the error detection. How might the authors explain the lack of improvement over task success rates?


**Summary Of Recommendation:**

The paper is a novel imitation learning method that is applied to mobile manipulation which comes with effective error detection mechanisms for out of distribution inputs. Future improvements could be made from clearer explanations of the significance of experimental results, choice of model architectures, and more insights into error correction strategies.

---

> ### Author Response · Authors · 2021-08-28
> **Response to Reviewer dB9j**
>
> Please note that our response has been split into two parts due to space constraints. (Part 1/2)
>
> Thank you for the helpful feedback and comments! We appreciate your patience in awaiting a response; we’ve conducted multiple follow-up experiments in response to feedback, and it has taken time to recruit people and conduct a user study on our MoMaRT system.
>
> We are glad to hear that you were impressed with our imitation learning results for mobile manipulation. We’d also like to apologize for our website link being down -- we were unaware that there were access restrictions placed when we originally published. We encourage you to (re-)visit our website (https://sites.google.com/view/il-for-mm/home) to see additional helpful supplemental material.
>
> We have updated our manuscript and corresponding supplementary materials (appendices); all modified / added words are highlighted in red for easy viewing.
>
> Below, we have tried to thoroughly respond to each of your questions and concerns. Please let us know if any information is lacking or unclear:
>
> - “On choice of model architectures: from the experimental results, it is not entirely clear to me how TieredRNN perform better than BC-RNN, and it seems that BC-RNN (which I assume to be a simpler structure) should already give quite impressive results. Is there any reason that the authors choose the more complex model structure of TieredRNN over simplicity? It would be better if there could be more explanations in the paper.”
>
> To re-iterate from our paper, because TieredRNN performed the best on the majority of tasks, we use this model for evaluation. We highlight that in the tasks where memory becomes more important (Table Setup from Dresser which requires searching the drawer and the Table Cleanup tasks requiring dumping trash at a randomized trash location), the TieredRNN performs better, suggesting that TieredRNN may show benefits in other tasks requiring temporal dependencies.
>
> However, we’d also like to provide some further context that might not be as readily apparent from our paper. For one, we note that to provide fair comparison, we utilized similar total parameter counts for our TieredRNN and RNN models. Thus, from a parameter standpoint, each model is equally complex. However, the connections between nodes are certainly more complex in the TieredRNN compared to the vanilla RNN model -- this is the proposed benefit! We found during preliminary hyperparameter sweeping that the TieredRNN scales better than the RNN model given similar total parameter counts; that is, further increasing the RNN model’s parameter count via the hidden dimension size generally resulted in policy degradation. Moreover, we found the TieredRNN model to also provide performance gains on previous iterations of tasks that were not included in this work. These empirical observations led us to deem the TieredRNN valuable enough to include as our final evaluation model, despite providing marginal benefits in this specific setting. We hope to continue iterating on this model in future work. We’ve added the above comments to our paper in Appendix A.7.
>
> - “On experimental results: According to Table 2, the error detection results are quite impressive, but they do not help success rate that much. In fact in many sections, the success rate decreases because of the error detection. How might the authors explain the lack of improvement over task success rates?”
>
> This is quite accurate. We apologize for the lack of clarity in our paper, and in addition to refining the wording in our paper, would like to clarify the intended role of our error detector. First, we want to highlight the distinction between error detection and error correction: Our method is not meant to (and cannot be expected to) improve task success (correction), given that it is not given additional online training data. However, our method can robustly distinguish failure states (detection) during online evaluation despite being fully trained offline using the same training data as the policy. This is the key benefit and intended contribution of our method, which we argue is especially useful for mobile manipulation because (a) common online error intervention methods (eg.: DAgger-based methods) are prohibitively expensive given the large state and action space and human supervision required, and (b) safety is paramount, where robots’ workspaces are dynamic and potentially unregulated. Please see the highlighted lines in 5.2 for the added wording, where we clarify that the success rates comparisons were mainly used to highlight that our error detector is not overly conservative and does not prematurely terminate episodes when there is no true error at hand.

---

> > ### Author Response · Authors · 2021-08-28
> > **Response (Part 2)**
> >
> > Please note that our response has been split into two parts due to space constraints. (Part 2/2)
> >
> > - “The current interventions when an error is detected are hardcoded programming if I interpret it correctly - only "recover" and "terminate". I'm curious of whether the authors have considered more versatile intervention methods, e.g. human monitoring and tele-operated corrections?”
> >
> > This is a great point! Error correction is an exciting field of research, and especially intriguing within the scope of mobile manipulation, where the distribution of errors can be much more broad compared to static manipulation or pure navigation. Your suggestions provide an avenue for future work that we’d like to explore more thoroughly in a focused manner! We did consider alternative options, but ultimately chose to proceed with our fully-offline error detector method due to its minimal (zero) additional training data requirements and our focus on error detection, rather than policy improvement. To be specific, DAgger and other human-in-the-loop methods are powerful at improving policy robustness, but comes at the cost of collecting additional online training samples. Human data is expensive, and this is especially compounded in mobile manipulation, where tasks are often long-horizon with wide state space coverage (as in our case). In contrast, our method requires no additional online samples, and is fully trained offline. We highlight that our method is not intended to consistently compensate for errors during intervention, but rather detect and terminate execution to prevent potentially dangerous out-of-distribution policy behavior from occurring that could damage the robot and / or its surroundings.
> >
> > We did evaluate our error detection mechanism against other fully-offline methods, to show that our method is in fact relatively more performant compared to obvious alternatives. In addition to our main reconstruction loss metric, we consider another loss metric (KL Divergence loss for the VAE model), latent metrics (VAE Encoder Variance Mean / Max values), and a policy uncertainty metric (policy action log probability). We find that our model strongly outperforms all of these methods, and is the only model able to achieve consistently high error detection precision and recall rates. This further validates prior work (https://arxiv.org/pdf/2012.00201.pdf, http://www.roboticsproceedings.org/rss13/p64.pdf, https://www.sciencedirect.com/science/article/pii/S016516841300515X ) that has found reconstruction error to be a viable metric for detecting errors, and also highlights the value of our method's ability to be easily tuned and robust across multiple tasks, which is a property not apparent in the other baselines. (Indeed, we found that for the latent and log probability methods, the error signal was very noisy and did not necessarily transfer well between tasks given the same threshold error value). Exact results and further analysis can be found in Appendix A.8.

---

> > > ### Comment · Reviewer_dB9j · 2021-09-03
> > > **Thank you for your response**
> > >
> > > Thank you authors for the detailed response. I think that the concerns in the reviews are further explained and clarified and I am leaning more towards acceptance.

---

### Official Review · Reviewer_3mTs · 2021-07-22

**Originality:** Very Good
**Technical Quality:** Fair
**Clarity Of Presentation:** Good
**Impact:** 4

**Recommendation:**

Strong Accept: I recommend accepting the paper and will argue for my recommendation even if other reviewers hold a different opinion.

**Summary:**

This paper proposed imitation learning (behavioral cloning (BC) more specifically) with an error detector by variational autoencoder (VAE).
In order to automate mobile manipulation, the authors first developed a teleoperation framework, MOMART, for collecting huge dataset, which is utilized for imitation learning.
Since mobile manipulation is a long-term operation, the effect of covariate shift, which is a shortcoming of BC, becomes more pronounced.
To prevent malfunctions caused by this, the authors additionally implemented the error detection mechanism based on the recovery accuracy of VAE.
Although it is unclear how high the success rate should be, the authors showed a relatively high percentage of error detection.

**Issues:**

- I think the error detection based on the reconstruction accuracy has a disadvantage that the threshold setting can be complicated depending on the observation information.
There are several existing methods based on latent variables, and I would like to see a comparison with such methods.
- The benefits of error detection to the task accomplishment were not sufficiently demonstrated.
I would like to see a comparison for indicators other than success rate (such as the load caused by the misoperation).
- The latent prior distribution and the output distribution of BC are assumed to be mixture distributions, but how do the authors give the number of mixtures? Isn't it task-dependent?
- What kind of observations are included in the constructed dataset?

**Reviewer Expertise:**

Very good: Comprehensive knowledge of the area

**Strengths And Weaknesses:**

Strengths:
- The authors collected a huge dataset of mobile manipulation that had not yet been constructed, although it was simulated.
- Focusing on that the observation of the dataset is from POMDP, the authors combined BC with TieredRNN to use the observation history appropriately, which makes the task successful with some probability.
- The VAE-based error detection mechanism achieved the high error detection rate even in the generalized setting.

Weaknesses:
- The newly introduced error detection mechanism has not improved the success rate of the task itself.
- Its implementation is also naive, and there is room for improvement.
- The authors have only verified the developed framework, and have not sufficiently examined and compared other possible implementations.

**Summary Of Recommendation:**

There is not much novelty in the technology, and it is not a good proposal because various patterns of error detection by VAE have been proposed so far.
On the other hand, the construction of a large dataset of long-horizon mobile manipulation tasks, which is closer to real-world problems, is very useful, and I commend the authors for this.
If qualitative or experimental comparisons on error detection are added, the paper seems to be worthy enough to be accepted.

---

> ### Author Response · Authors · 2021-08-28
> **Response to Reviewer 3mTs**
>
> Please note that our response has been split into three parts due to space constraints. (Part 1/3)
>
> Thank you for the useful questions and feedback! We appreciate your patience in awaiting a response; we’ve conducted multiple follow-up experiments in response to feedback, and it has taken time to recruit people and conduct a user study on our MoMaRT system.
>
> We have updated our manuscript and corresponding supplementary materials (appendices); all modified / added words are highlighted in red for easy viewing.
>
> We are happy that you think our dataset will be beneficial to the robot learning community. We’d also like to apologize for our website link being down -- we were unaware that there were access restrictions placed when we originally published. We encourage you to (re-)visit our website (https://sites.google.com/view/il-for-mm/home) to see additional helpful supplemental material.  You brought up some great points, and we’ve tried to thoroughly respond below. Please let us know if any information is lacking or unclear:
>
> - “The newly introduced error detection mechanism has not improved the success rate of the task itself.”
>
> This is a great point -- and we completely agree! We apologize for the lack of clarity in our paper, and in addition to refining the wording in our paper, would like to clarify the intended role of our error detector. First, we want to highlight the distinction between error detection and error correction: Our method is not meant to (and cannot be expected to) improve task success (correction), given that it is not given additional online training data. However, our method can robustly distinguish failure states (detection) during online evaluation despite being fully trained offline using the same training data as the policy. This is the key benefit and intended contribution of our method, which we argue is especially useful for mobile manipulation because (a) common online error intervention methods (eg.: DAgger-based methods) are prohibitively expensive given the large state and action space and human supervision required, and (b) safety is paramount, where robots’ workspaces are dynamic and potentially unregulated. Please see the highlighted lines in 5.2 for the added wording, where we clarify that the success rates comparisons were mainly used to highlight that our error detector is not overly conservative and does not prematurely terminate episodes when there is no true error at hand.
>
>
> - “Its implementation is also naive, and there is room for improvement.”
>
> We agree that our method is simple -- there is little novelty from the modeling perspective. With regard to our error detection implementation being simple, we follow prior work that has shown this to be an effective method (https://arxiv.org/pdf/2012.00201.pdf, http://www.roboticsproceedings.org/rss13/p64.pdf, https://www.sciencedirect.com/science/article/pii/S016516841300515X ). Within this context, our method generally performs surprisingly well with very high precision and recall rates, across all of our tasks and over multiple types of data (expert, suboptimal, few-shot generalize). Thus, while simple in form, our error detector is shown to be effective despite being trained using zero additional data and can be expected to reliably perform and mitigate potentially dangerous erroneous policy behavior from occurring. This is the main takeaway that may have not been conveyed as explicitly as it should have been, and we’ve improved the wording in our paper to make this more clear.

---

> > ### Author Response · Authors · 2021-08-28
> > **Response (Part 2)**
> >
> > Please note that our response has been split into three parts due to space constraints. (Part 2/3)
> >
> > - “The authors have only verified the developed framework, and have not sufficiently examined and compared other possible implementations.”
> >
> > This is a valid issue, and we would like to respond by first distinguishing between comparisons with other interface modalities and other interface implementations. With respect to other interface modalities (VR, physical controllers, etc.), MoMaRT’s key advantage is scalability -- no additional hardware is needed, other than a smartphone (iPhone) and a laptop / desktop, which allows demonstrations to be easily and rapidly collected from anywhere in the world. Direct comparison with these other methods is outside the scope of this work.
> >
> > With respect to other possible implementations, we had iterated significantly on multiple paradigms with 5 users with varying levels of teleoperation experience (from novice to expert) providing feedback, and converged to the proposed interface. While there is pre-existing work that has already conducted user studies on the original RoboTurk interface which our platform is built upon (https://arxiv.org/pdf/1911.04052.pdf) that showcases its efficacy, we conducted a small-scale user study with 10 new users to provide qualitative and quantitative evaluation of our platform. In general, users were usually able to feel comfortable with the interface within 15 minutes, but struggled to confidently teleoperate the mobile manipulator, though users were only allotted 30 minutes of playtesting time. Much of the user feedback highlighted the difficulties associated with the POMDP mobile manipulation setting, such as limited field of view and difficulty executing precise arm trajectories to operate articulated objects. All users found the arm reset functionality to be beneficial during teleoperation, validating this key design decision in our interface. Despite the challenges faced with our system and difficult domain, the merits are clear in the ultimate scale of data and learning results we have already been able to produce, and hope that future work can continue to build upon our platform to increase its efficacy for users. We encourage you to see the full details and discussion of results of our user study in Appendix A.5.
> >
> >
> > - “I think the error detection based on the reconstruction accuracy has a disadvantage that the threshold setting can be complicated depending on the observation information. There are several existing methods based on latent variables, and I would like to see a comparison with such methods.”
> >
> > While generally we would agree that manual tuning can be complicated, we found that in our setting this was not the case, and even unnecessary. To be clear: there was no per-evaluation tuning of this error setting in order to achieve our high precision and recall rates, this resulted solely from a single setting applied across all evaluations. Indeed, we only briefly tuned our error threshold setting based on a few observed rollouts, and then proceeded to set this as a constant value across all tasks and all evaluation settings (expert, suboptimal, few-shot generalization). While this may be specific to our specific environment (kitchen) setting, at the very least this suggests that a single setting is robust to a wide distribution of errors with varying visuals, and may be useful in other similar domains as well.
> >
> > With regards to latent variable baselines, we conducted an additional experiment comparing our error detector mechanism to other alternatives. In addition to our main reconstruction loss metric, we consider another loss metric (KL Divergence loss for the VAE model), latent metrics (VAE Encoder Variance Mean / Max values), and a policy uncertainty metric (policy action log probability). We find that our model strongly outperforms all of these methods, and is the only model able to achieve consistently high error detection precision and recall rates. This further validates the prior work that has found reconstruction error to be a viable metric for detecting errors, and also highlights the value of our method's ability to be easily tuned and robust across multiple tasks, which is a property not apparent in the other baselines. (Indeed, we found that for the latent and log probability methods, the error signal was very noisy and did not necessarily transfer well between tasks given the same threshold error value). Exact results and further analysis can be found in Appendix A.8.

---

> > > ### Author Response · Authors · 2021-08-28
> > > **Response (Part 3)**
> > >
> > > Please note that our response has been split into three parts due to space constraints. (Part 3/3)
> > >
> > > - “The latent prior distribution and the output distribution of BC are assumed to be mixture distributions, but how do the authors give the number of mixtures? Isn't it task-dependent?”
> > >
> > > The hyperparameters were tuned for one task, and then applied to all five tasks during our final evaluation. We use a mixture distribution of 5 in all cases. In general, using a GMM vs. using a single-mode policy resulted in a larger policy gain compared to using different numbers of modes. We include the major hyperparameter choices in Appendix A.3.
> > >
> > >
> > > - “What kind of observations are included in the constructed dataset?”
> > >
> > > Our dataset consists of head and wrist RGB-D frames, proprioceptive states, and LIDAR range scans. We also include ground-truth simulation states for playing back trajectories.

---

> > > > ### Comment · Reviewer_3mTs · 2021-09-02
> > > > **Thank you for your response**
> > > >
> > > > I thank the authors for addressing my comments. I believe that the manuscript is suitable for acceptance.

---

### Official Review · Reviewer_56kE · 2021-07-29

**Originality:** Excellent
**Technical Quality:** Very Good
**Clarity Of Presentation:** Excellent
**Impact:** 4

**Recommendation:**

Strong Accept: I recommend accepting the paper and will argue for my recommendation even if other reviewers hold a different opinion.

**Summary:**

This paper describes a new method for collecting demonstrations for long-horizon tasks in mobile manipulation. The approach manages the gap between demonstrator observations and the systems observations. The authors use this to collect a large dataset of long-horizon demonstrations and achieve state-of-the-art results by imitating them. The authors also propose an error detection method to deal with the potentially large shift in states observed between the expert demonstrations and the learned policy.

**Issues:**

In my read through the paper, I did not see discussion of publishing the dataset of demonstrations. Can the authors please clarify how they will share this resource (or point me to the place in the paper where I have missed it)?

Please address the minor comments.

Please provide more analysis of the few-shot results.

Consider adding evaluations of the design choices in the error detection method.

**Reviewer Expertise:**

Excellent: Expert knowledge on the topic of the paper

**Strengths And Weaknesses:**

Strengths
--------------

The paper is well-written and gives a good characterization of related work.

The demonstration collection method looks novel, non-trivial, and effective. The dataset collected looks like it could be quite useful to the community.

The policies the authors train go beyond the state-of-the-art in policy learning for long-horizon tasks. This is an important area for the community and the results are impressive.

Weaknesses
-------------------

While the error detection method looks to be useful and clearly helps with imitating the data, the error detection method itself is not really evaluated. In particular, the authors could evaluate their design choices for the error detection. This is important if the authors wish to claim the error detection method as a contribution.

I found the discussion of the few-shot generalization results to be too short. I recommend the authors expand this section of the text (l.307-322). In particular, including some details about the drop in performance to reduce the burden on the reader.

Minor Comments
------------------------
[l.173] "110 successful demonstration per from" -- missing word
[l.188] in a POMDP, $\pi$ is a function of the full history, not just the last observation
[l.240] "Algorithm 1 formalizes this method" --- the paper does not appear to have an algorithm 1



**Summary Of Recommendation:**

As my review indicates, I'm a big fan of this paper. It makes an important step forward and contributes resources that will help the community progress. I will advocate for the paper's acceptance, if only selfishly so that I can build on the dataset and tools they have provided.


Update after rebuttal
----------------------------
I thank the authors for addressing my comments and I am happy they have included more discussion of the error detection method they propose.

---

> ### Author Response · Authors · 2021-08-28
> **Response to Reviewer 56kE**
>
> Please note that our response has been split into two parts due to space constraints. (Part 1/2)
>
> Thank you for the helpful feedback and comments! We appreciate your patience in awaiting a response; we’ve conducted multiple follow-up experiments in response to feedback, and it has taken time to recruit people and conduct a user study on our MoMaRT system.
>
> We have updated our manuscript and corresponding supplementary materials (appendices); all modified / added words are highlighted in red for easy viewing.
>
> We are excited that others such as yourself are already eager to leverage our data to further explore mobile manipulation. We’d also like to apologize for our website link being down -- we were unaware that there were access restrictions placed when we originally published. We encourage you to (re-)visit our website (https://sites.google.com/view/il-for-mm/home) to see additional helpful supplemental material.
>
> Below, we have tried to thoroughly respond to each of your questions and concerns. Please let us know if any information is lacking or unclear:
>
> - “While the error detection method looks to be useful and clearly helps with imitating the data, the error detection method itself is not really evaluated. In particular, the authors could evaluate their design choices for the error detection. This is important if the authors wish to claim the error detection method as a contribution.”
>
> This is a great point! We’d like to first clarify that we are not claiming that the error detector itself is a contribution, but rather the strong empirical resulting from our end-to-end learning pipeline in the complex mobile manipulation setting (>45% success and >85% error detector precision and recall across all tasks). However, while we have extensively evaluated our error detector on a wide variety of conditions (expert data, suboptimal data, out-of-distribution data), we can better contextualize the efficacy of our method by evaluating alternative baselines. In addition to our main reconstruction loss metric, we consider another loss metric (KL Divergence loss for the VAE model), latent metrics (VAE Encoder Variance Mean / Max values), and a policy uncertainty metric (policy action log probability). We find that our model strongly outperforms all of these methods, and is the only model able to achieve consistently high error detection precision and recall rates. This further validates prior work (https://arxiv.org/pdf/2012.00201.pdf, http://www.roboticsproceedings.org/rss13/p64.pdf, https://www.sciencedirect.com/science/article/pii/S016516841300515X ) that has found reconstruction error to be a viable metric for detecting errors, and also highlights the value of our method's ability to be easily tuned and robust across multiple tasks, which is a property not apparent in the other baselines. (Indeed, we found that for the latent and log probability methods, the error signal was very noisy and did not necessarily transfer well between tasks given the same threshold error value). Exact results and further analysis can be found in Appendix A.8.
>
>
> - “I found the discussion of the few-shot generalization results to be too short. I recommend the authors expand this section of the text (l.307-322). In particular, including some details about the drop in performance to reduce the burden on the reader.”
>
> Thank you for this feedback! We acknowledge that we have omitted explicit interpretations of the drop in policy performance under few-shot generalization settings. Transfer learning is a difficult task, and becomes more pronounced in settings where the distribution shift is not marginal and continuous (e.g.: wider distribution of start states) but rather significant and discrete (e.g.: our setting, with key object locations swapped), and makes policy generalization challenging in our long-horizon task setup requiring continuous visuo-motor policy execution. Under these settings, it is expected that policy performance will drop without sufficient amounts of data in the new setting.
>
> However, this raises a key question: to show that our finetuning generalization scheme is actually beneficial, we must compare against training purely on the domain-shifted, few-shot demonstrations. We evaluated this baseline, and found these policies to all perform worse (Appendix A.9)), showing that the amount of information transfer from the original task training distribution is non-negligible and provides concrete policy improvement.

---

> > ### Author Response · Authors · 2021-08-28
> > **Response (Part 2)**
> >
> > Please note that our response has been split into two parts due to space constraints. (Part 2/2)
> >
> > - "I did not see discussion of publishing the dataset of demonstrations. Can the authors please clarify how they will share this resource (or point me to the place in the paper where I have missed it)?"
> >
> > Thank you for pointing this out. Upon acceptance, we plan to release the dataset in a compatible form with other off the shelf platforms along with our codebase for easy experiment reproduction and rapid prototyping. The initial dataset and code release will be posted to our paper’s website: https://sites.google.com/view/il-for-mm/home.
> >
> > Minor comments addressed:
> >
> > > [l.173] "110 successful demonstration per from" -- missing word
> >
> > Fixed -- added “task”
> >
> > > [l.188] in a POMDP, π  is a function of the full history, not just the last observation
> >
> > This has been updated
> >
> > > [l.240] "Algorithm 1 formalizes this method" --- the paper does not appear to have an algorithm 1
> >
> > We apologize for this confusion -- Algorithm 1 can be found in Appendix A.2

---

### Meta-Review · Area_Chair_gAtd · 2021-08-14

**Recommendation:** Accept (Poster)
**Confidence:** 4

**Metareview:**

**After rebuttal**
I thank the authors for their thorough responses, my concerns have been addressed, and the reviewers concerns seems to have been addressed to the reviewers satisfaction.
Thus I recommend accept.

**Initial Meta-review**
**Summary**

This work presents a teleoperation system for long horizon tasks performed by mobile manipulation robots. This teleoperation framework is utilized to collect a continuous control dataset in a simulated kitchen domain consisting of over 1200 successful demonstrations across five long-horizon tasks.  The author then present initial imitation learning results on this dataset. The imitation learning process is augmented with a learned error detector which can detect when the agent is in a failure state.

**Strengths**:
This work contributes a system for mobile manipulation data collection, which is an important and active research are in the robotics community

The manuscript is well written and the work is well motivated

The work introduces a variant of behavior cloning that leverages hierarchical abstractions, and show that imitation learning with this variant outperforms some relevant baselines

**Weaknesses**:

This work also introduces an error detection method, however it is barely evaluated.
All reviewers are missing some more detailed analysis and discussion on various aspects of the evaluation

---

> ### Author Response · Authors · 2021-08-30
> **Response to Area Chair gAtd**
>
> Thank you for taking the time to review our work and summarize the reviewers' feedback! We appreciate your patience in awaiting a response; we’ve conducted multiple follow-up experiments in response to feedback, and it has taken time to recruit people and conduct a user study on our MoMaRT system.
>
> We have updated our manuscript and corresponding supplementary materials (appendices); all modified / added words are highlighted in red for easy viewing.
>
> Below, we summarize the key questions reviewers had about our error detector, and respond in detail to each. Please let us know if there is any confusion or if further clarification is needed:
>
> > Error Detector does not seem to be fully evaluated, and is a weak contribution.
>
> We’d like to first clarify that we are not claiming that the error detector itself is a contribution, but rather the strong empirical resulting from our end-to-end learning pipeline in the complex mobile manipulation setting (>45% success and >85% error detector precision and recall across all tasks). However, while we have extensively evaluated our error detector on a wide variety of conditions (expert data, suboptimal data, out-of-distribution data), we can better contextualize the efficacy of our method by evaluating alternative baselines. In addition to our main reconstruction loss metric, we consider another loss metric (KL Divergence loss for the VAE model), latent metrics (VAE Encoder Variance Mean / Max values), and a policy uncertainty metric (policy action log probability). We find that our model strongly outperforms all of these methods, and is the only model able to achieve consistently high error detection precision and recall rates. This further validates prior work (https://arxiv.org/pdf/2012.00201.pdf, http://www.roboticsproceedings.org/rss13/p64.pdf, https://www.sciencedirect.com/science/article/pii/S016516841300515X ) that has found reconstruction error to be a viable metric for detecting errors, and also highlights the value of our method's ability to be easily tuned and robust across multiple tasks, which is a property not apparent in the other baselines. (Indeed, we found that for the latent and log probability methods, the error signal was very noisy and did not necessarily transfer well between tasks given the same threshold error value). Exact results and further analysis can be found in Appendix A.8.
>
> > Error Detector does not seem to increase success rate.
>
> We apologize for the lack of clarity in our paper, and in addition to refining the wording in our paper, would like to clarify the intended role of our error detector. First, we want to highlight the distinction between error detection and error correction: Our method is not meant to (and cannot be expected to) improve task success (correction), given that it is not given additional online training data. However, our method can robustly distinguish failure states (detection) during online evaluation despite being fully trained offline using the same training data as the policy. This is the key benefit and intended contribution of our method, which we argue is especially useful for mobile manipulation because (a) common online error intervention methods (eg.: DAgger-based methods) are prohibitively expensive given the large state and action space and human supervision required, and (b) safety is paramount, where robots’ workspaces are dynamic and potentially unregulated. Please see the highlighted lines in 5.2 for the added wording, where we clarify that the success rates comparisons were mainly used to highlight that our error detector is not overly conservative and does not prematurely terminate episodes when there is no true error at hand.
>
> > There are much more effective options for error compensation (e.g.: DAgger)
>
> We did consider alternative options, but ultimately chose to proceed with our fully-offline error detector method due to its minimal (zero) additional training data requirements and our focus on error detection, rather than policy improvement. To be specific, DAgger and other human-in-the-loop methods are powerful at improving policy robustness, but comes at the cost of collecting additional online training samples. Human data is expensive, and this is especially compounded in mobile manipulation, where tasks are often long-horizon with wide state space coverage (as in our case). In contrast, our method requires no additional online samples, and is fully trained offline. We highlight that our method is not intended to consistently compensate for errors during intervention, but rather detect and terminate execution to prevent potentially dangerous out-of-distribution policy behavior from occurring that could damage the robot and / or its surroundings.

---

### Decision · Program_Chairs · 2021-09-13

**Decision:**

Accept (Poster)

**Comment:**

**After rebuttal**
I thank the authors for their thorough responses, my concerns have been addressed, and the reviewers concerns seems to have been addressed to the reviewers satisfaction.
Thus I recommend accept.

**Initial Meta-review**
**Summary**

This work presents a teleoperation system for long horizon tasks performed by mobile manipulation robots. This teleoperation framework is utilized to collect a continuous control dataset in a simulated kitchen domain consisting of over 1200 successful demonstrations across five long-horizon tasks.  The author then present initial imitation learning results on this dataset. The imitation learning process is augmented with a learned error detector which can detect when the agent is in a failure state.

**Strengths**:
This work contributes a system for mobile manipulation data collection, which is an important and active research are in the robotics community

The manuscript is well written and the work is well motivated

The work introduces a variant of behavior cloning that leverages hierarchical abstractions, and show that imitation learning with this variant outperforms some relevant baselines

**Weaknesses**:

This work also introduces an error detection method, however it is barely evaluated.
All reviewers are missing some more detailed analysis and discussion on various aspects of the evaluation